# A Clustering-Driven Approach to Predict the Traffic Load of Mobile Networks for the Analysis of Base Stations Deployment

**Basma Mahdy** [1], **Hazem Abbas** [2], **Hossam S. Hassanein** [2,*], **Aboelmagd Noureldin** [3] and **Hatem Abou-zeid** [4]

1.   Department of Electrical and Computer Engineering, Queen's University, Kingston, ON K7L 3N6, Canada; basma.mahdy@queensu.ca
2.   School of Computing, Queen's University, Kingston, ON K7L 3N6, Canada; hazem.abbas@queensu.ca
3.   Department of Electrical and Computer Engineering, Royal Military College of Canada, Kingston, ON K7K 7B4, Canada; Aboelmagd.Noureldin@rmc.ca
4.   Ericsson Canada, Ottawa, ON K2K 2V6, Canada; hatem.abou-zeid@ericsson.com
*   Correspondence: hossam@cs.queensu.ca

**Abstract:** Mobile network traffic is increasing in an unprecedented manner, resulting in growing demand from network operators to deploy more base stations able to serve more devices while maintaining a satisfactory level of service quality. Base stations are considered the leading energy consumer in network infrastructure; consequently, increasing the number of base stations will increase power consumption. By predicting the traffic load on base stations, network optimization techniques can be applied to decrease energy consumption. This research explores different machine learning and statistical methods capable of predicting traffic load on base stations. These methods are examined on a public dataset that provides records of traffic loads of several base stations over the span of one week. Because of the limited number of records in the dataset for each base station, different base stations are grouped while building the prediction model. Due to the different behavior of the base stations, forecasting the traffic load of multiple base stations together becomes challenging. The proposed solution involves clustering the base stations according to their behavior and forecasting the load on the base stations in each cluster individually. Clustering the time series data according to their behavior mitigates the dissimilar behavior problem of the time series when they are trained together. Our findings demonstrate that predictions based on deep recurrent neural networks perform better than other forecasting techniques.

**Keywords:** 5G networks; wireless communications; network load predictions; clustering

## 1. Introduction

Over the last few years, the traffic load on wireless networks has been increasing exponentially. The Ericsson Mobility Report stated that over the past two years (2018–2019), mobile data traffic increased by 49%, reaching up to 40 Exabytes [1]. Although it is generally agreed that the increasing amount of data used by smartphone users is a major factor for this growth, the introduction of data usage in many smart devices is another reason for this rapid growth. One of the consequences of the enormous growth in data usage is the need for more base stations in the network to serve more devices. However, when more base stations are added, a significant increase in energy consumption occurs. Base stations require energy equal to approximately 70% of the total energy of the entire network infrastructure. With the ability to predict the traffic on base stations, the opportunity to save energy arises by developing a method to switch off base stations during off-load times. Traffic in base

stations fluctuates constantly; therefore, it is impracticable to assume that off-peak hours are always constant. Traffic predictions can be used to selectively put some base stations into sleep mode during varying off-peak hours while ensuring that the service will still be provided by the active base stations. Multiple works have been conducted to study the trade-off between energy consumption and the quality of service (QoS) with base station sleeping techniques [2,3].

Various energy-saving techniques are being considered for 5G cellular networks. 5G cellular networks were developed as a response to the drastic rise in users' demands and the exponential growth of mobile data traffic. The latest Ericsson report forecasts 190 million 5G subscribers worldwide by the end of 2020 and 2.8 billion by 2025 [4]. This enormous growth in subscribers will lead to a dramatic increase in mobile data traffic volume, which will require a dramatic increase in the provided capacity compared to previous cellular systems [5,6].

Several techniques have been introduced in the literature to achieve this increase in capacity without the need to increase energy consumption [5,6]. For instance, optimal resource allocation techniques have been used with optimization methods and mathematical tools to allocate radio resources where energy consumption is minimized [5–7]. Different optimal network deployment techniques, such as dynamic base station sleeping mechanisms, adaptive load variation algorithms, and reducing cells' size, have been conducted to deploy the cellular network while ensuring that maximum throughput is achieved with a minimal amount of energy consumed [5–8]. Some techniques apply different hardware solutions, such as the cloud-based implementation of the radio access network and network function virtualization [5–9].

This research is centered around the problem of network traffic load prediction on base stations while focusing mainly on exploring the most dependable and accurate method for forecasting traffic load. Different base stations process different amounts of load traffic; some base stations within a network may have a low load, whereas other base stations may have an extremely high load. At any given time, the opposite could be true. For time series forecasting, the larger the training data, the better the performance one can achieve. In the case of the data encountering different behaviors, this may not improve performance and possibly degrade it. Hence, base stations with similar traffic loads need to be trained together. Clustering was used in this paper to group base stations with similar traffic loads to solve this problem.

In the literature, predicting the load on base stations was addressed multiple times for energy saving. The authors of ref. [10] used base stations' load prediction to create a base station sleeping mechanism. The output of the prediction was used to determine if the load would be low. If the base station encountered low load, it would switch off, and the base station's users would be switched to Pico base stations (PBSs) from macro base stations (MBSs) [10]. Simulation results demonstrated that PBSs require less power than MBSs, and this methodology reduced the energy consumption in the network. An additional energy-saving schema was proposed by the authors of ref. [11]. The future load prediction on these base stations was used to develop a grid-based energy-saving scheme. The authors claimed that base stations with low loads can be switched to sleep mode without degradation in the quality of experience. In this schema, the authors took advantage of the overlapping coverage areas between neighboring base stations. The schema developed considers the future prediction and checks if any base stations are overlapping. The extra base stations will then be switched off until the future prediction shows they are needed, whereupon they will be switched back on.

Load forecasting processes have been applied in several ways. The most common method is using statistical methods such as the simple moving average (SMA), exponential smoothing (ES), and the autoregressive integrated moving average (ARIMA). Research has recently moved toward machine learning and deep learning methods and, most notably, recurrent neural networks to perform load forecasting. The authors of ref. [12] compared the performance of the ARIMA and exponential smoothing models for predicting load on base stations. The two models were examined using two scenarios. In the first scenario, the entire area was divided into multiple regions, and the loads on all the base stations were predicted in each region together. In the second scenario, the load predictions

on each base station were examined individually. From the two scenarios, the ARIMA model provided lower errors in predicting load during the weekdays; however, the exponential smoothing model was better at predicting load on the weekends and single cells [13]. Machine learning was used to perform the prediction process [13], where real past records were utilized to create a support vector regression (SVR) model that performs the prediction. The authors claimed that they achieved superior performance even with the change in load behavior.

The research aims to assess the most popular time series forecasting techniques to predict the traffic load on base stations within a permissible margin of error, outlined in the following four steps:

1. Selecting a public dataset that can be used for the task of forecasting the traffic load of base stations.
2. Analyzing the selected dataset to extract useful information and applying the required data preprocessing techniques on it.
3. Clustering the base stations according to their behavior before applying the forecasting techniques and examining the effect of clustering on the accuracy of the forecasting results.
4. Comparing the performance of commonly used time series forecasting techniques (statistical, machine learning, and deep learning methods) and how they perform in forecasting the traffic load of base stations.

## 2. Data Preprocessing

The dataset used in this research is the City Cellular Traffic Map (C2TM) [14], collected in China for a cellular area of 50 km × 60 km during the period from 19 August 2012 to 26 August 2012. The dataset covers the hourly traffic load on 13,296 base stations for seven days or 168 h. The Georgian calendar is used for the representation of time. For the location, the longitude and latitude of the connected base stations are reported in the dataset. Due to privacy reasons, the locations used in this dataset are not the real locations of the base stations but rather meshed locations [14].

Figure 1 illustrates the average load on a few base stations during the seven-day period. The load clearly differs from one base station to the other; however, there is a general upward trend between 10 h and 15 h. The load expectedly decreases on most of the base stations after 18 h until 23 h because the load during rush hours and the daytime is usually higher than the load at the late hours of the night.

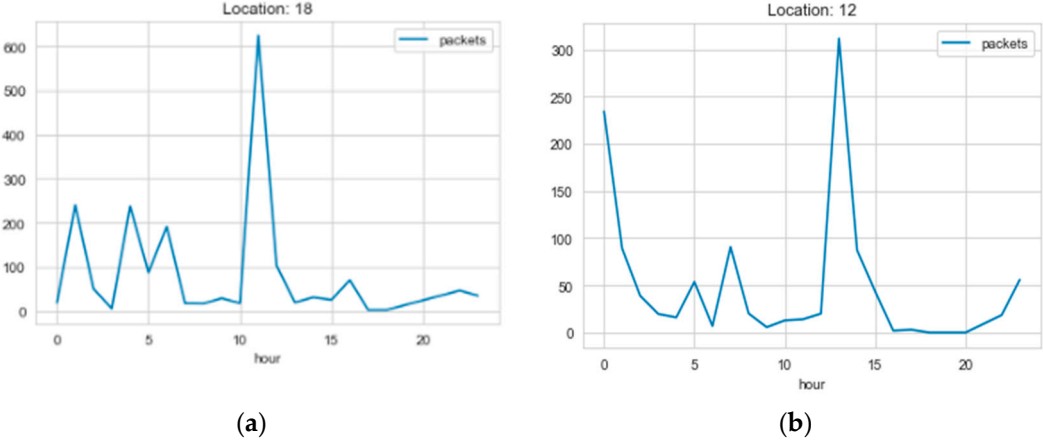

(**a**)　　　　　　　　　　　　　　　　　　　　　(**b**)

**Figure 1.** Average weekly load on different base stations: (**a**) location 18; (**b**) location 12.

Most time series forecasting techniques require the time series to be stationary. Stationarity indicates that the statistical properties of the time series remain constant over time. For a time series to be stationary, it needs both the standard deviation and the mean to not change over time. The rolling mean and the rolling standard deviation of the load of each base station over the seven days were calculated to test the stationarity of the load of the base stations. It was observed that the mean and standard deviation of the time series in the dataset were not constant over time, demonstrating that

the load of the base stations was not stationary. Figure 2 illustrates the load of a sample base station and displays the rolling standard deviation and rolling mean through the week. Hence, the dataset required additional transformation to convert it to a stationary time series.

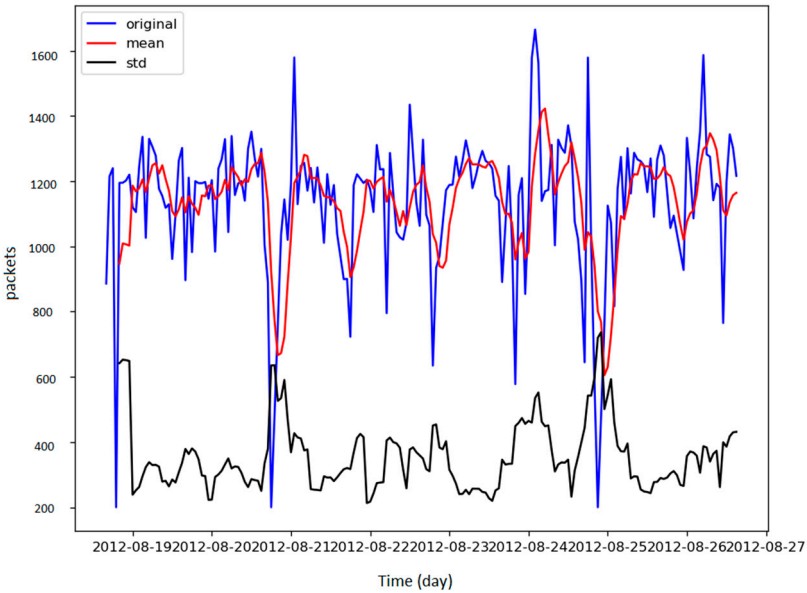

**Figure 2.** Standard deviation and mean vs. the load.

To properly use the data for forecasting, several data preparation steps were applied to the dataset. The first step in the data preparation process is data cleaning, comprising four steps:

### 2.1. Removing Base Stations with Few Data Records

Although the dataset includes roughly 13,000 base stations, some of these base stations provided only few records in the whole dataset. A threshold was determined to represent the minimum number of records that the base station contributes in order to be considered. The base stations that did not satisfy this threshold were removed. The data were grouped by the base station ID. For each base station, the number of records it contained was counted. Different values were used to determine the best threshold for removing undesirable records. The best value was determined based on the total number of base stations and the average number of records for each base station. Accordingly, a threshold of 67, representing a value of 40% of the maximum number of records (24 h × 7 days = 168 h), was set. Only base stations with the number of records exceeding this threshold (67) were kept. The rest were removed.

### 2.2. Handling Outliers

Outliers are the data points distinctly different from the rest of the data points in the dataset. Outliers may exist in the dataset because of noise or other random measurement errors that may occur during the recording or processing of the data. The presence of these outliers in the data causes inconsistencies that can decrease the performance of the forecasting models. Hence, it is important to identify and remove the outliers before applying other steps. A box plot and scatter plot were used for the visualization of the outliers. The distribution of data before removing the outliers is illustrated in Figures 3 and 4 using the box plot and scatter plot, respectively. The figures demonstrate that a significant number of outliers were found in the dataset.

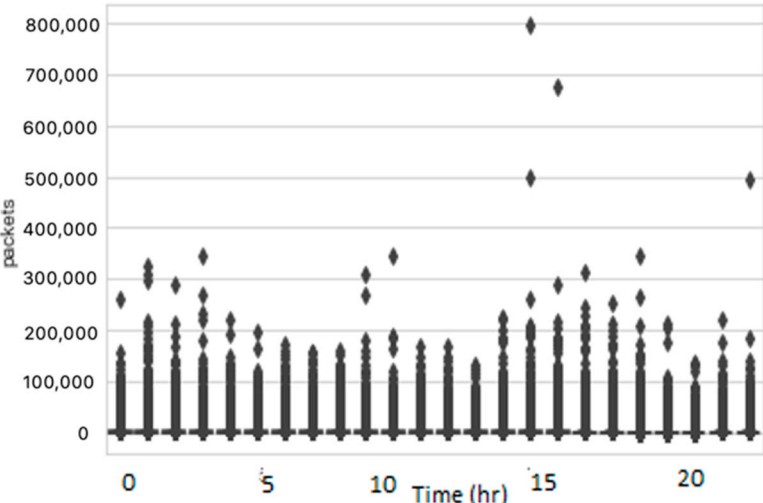

**Figure 3.** Box plot with outliers.

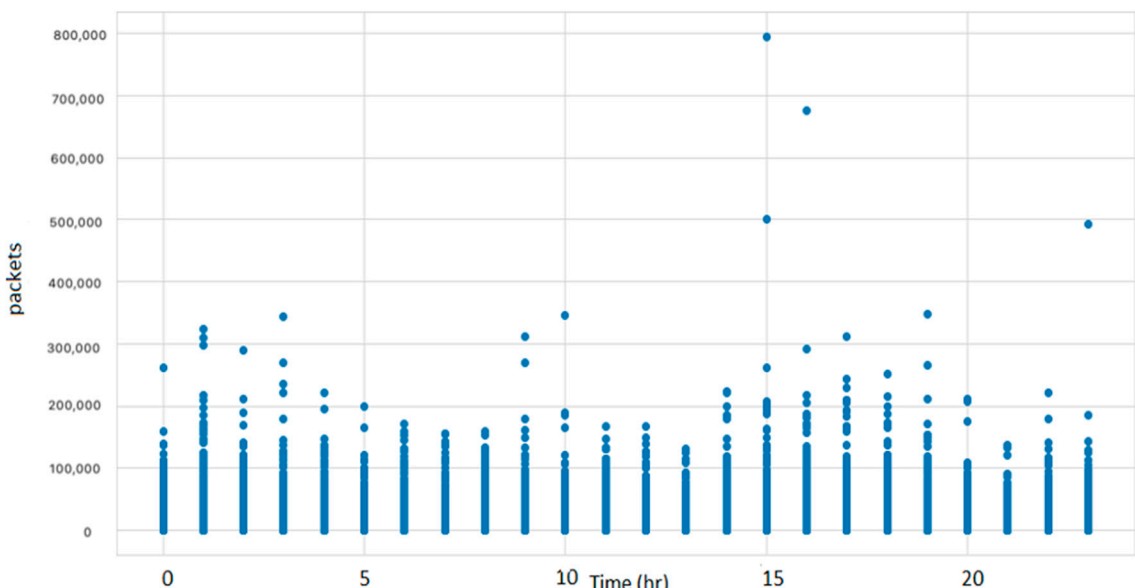

**Figure 4.** Scatter plot with outliers.

The Z-score method [15] was used to define the outliers in the data. The Z-score of all the data points was calculated. According to the empirical rule [16], 99.7% of the data points lie between a ±3 standard deviation. Hence, the interpretation of a datapoint that results in a Z-score greater than three is that this datapoint is different from 99.7% of the other data points. Therefore, a threshold of three was determined. For each data point, if the Z-score of the data point exceeded this threshold, the outlier value was initially replaced with a NaN, which is used to represent the missing values.

*2.3. Replacing NaNs*

Finally, for all the NaN values in the dataset that originated from the missing values or the outliers, we replaced this value with the value of the record at the previous time instance. As illustrated in the box plot in Figure 5 and the scatter plot in Figure 6, the number of outliers was significantly reduced.

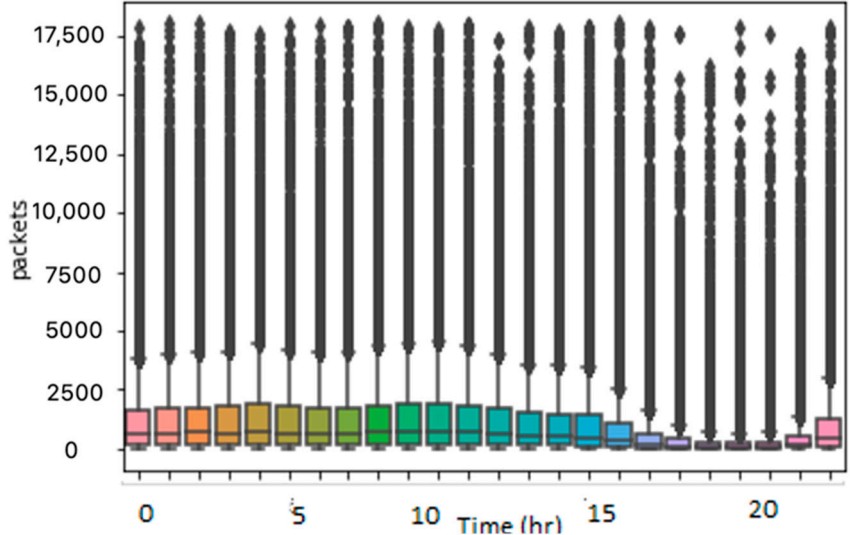

**Figure 5.** Box plot without outliers.

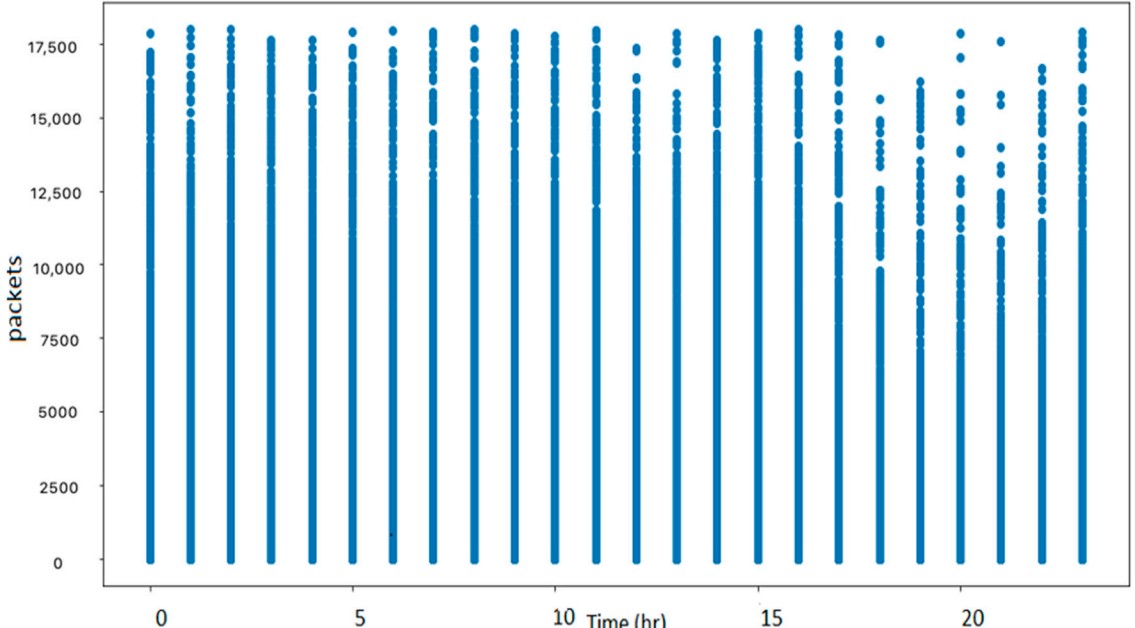

**Figure 6.** Scatter plot without outliers.

### 2.4. Feature Engineering

In this step, some new fields will be created from the existing ones. For instance, the dataset provides one field that contains the hourly timestamp in UNIX epoch time, from which the hour, day, and month fields were derived to test their individual effect on the base station load. A field, week_day, which determines if this day was a weekday or weekend, was also added. Lastly, a field that combines the longitude and latitude was derived using the Haversine distance, which uses the longitude and latitude coordinates of two points lying on a sphere to calculate the great-circle distance between them. The origin point of the distance coordinates was assumed to be $(\varphi, \lambda) = (0, 0)$, where $\varphi$ represents the latitude and $\lambda$ represents the longitude. The Haversine distance [17] between the base station

location's coordinates and the origin point was then measured. The distance s added to the dataset as a new field, haversine_dist, and calculated as:

$$d = 2r \arcsin\left(\sqrt{h}\right)$$
$$h = \sin^2\left(\frac{\varphi 1 - \varphi 2}{2}\right) + \cos(\varphi 1)\cos(\varphi 2)\left(\sin^2\left(\frac{\lambda 2 - \lambda 1}{2}\right)\right) \tag{1}$$

where $0 \le h \le 1$, and

- $h$ is the Haversine function;
- $d$ is the Haversine distance;
- $\varphi 1$ and $\lambda 1$ are the latitude and longitude of the origin point;
- $\varphi 2$ and $\lambda 2$ are the latitude and longitude of the base station.

### 2.4.1. Data Normalization

The data values have different scales. A normalization step was required to make all the data fields have the same scale. Normalization to a range of 0 to 1 was applied to all fields except for the hour field. To preserve its cyclical feature, it was transformed into two dimensions using sine and cosine transformation as follows:

- hour_sin = sin(*hour*);
- hour_cos = cos(*hour*).

### 2.4.2. Stationary Time Series

As previously mentioned, the data miss the stationarity characteristic. To convert the time series to a stationary time series, differencing, seasonal differencing, and log transformation techniques were examined. Differencing eliminates the level fluctuations in a time series, leading to a decrease in trend and seasonality. Hence, the mean and standard deviation can be kept stable [18]. The seasonal difference represents the difference between two consecutive observations lying in the same season [19]. For series S and a season with 24 periods, the seasonal differencing at period $t$ is $S_t - S_{t-24}$. Hence, in seasonal differencing, the observation taken at a specific hour was subtracted from the same hour on the previous day. Log transformation is a common transformation technique that works by applying the log function on the time series to transfer each value to its logged one. After using these three methods to convert the time series to stationary, log transformation was performed because it was more interpretable than the other two methods.

## 3. Clustering of Base Stations

As previously mentioned, clustering was used in this work to group the base stations with similar traffic loads together. Clustering time series data differs from clustering static data in composing dynamic values for the features [20]. Most traditional clustering techniques fail to provide satisfying results in clustering time series data because they are designed for extracting similarities from static features [21].

Clustering can be performed according to several properties. In this research, two properties were used to perform clustering.

1. Spatial clustering, which focuses on clustering the time series according to their location.
2. Time series clustering, which focuses on clustering the time series according to their behavior.

Spatial clustering assumes that adjacent base stations may contend with the same behavior. Figure 7 illustrates the locations of all the base stations. The map was divided into small grids where each grid represented a cluster. Base stations located in the same grid were assigned to the grid cluster number.

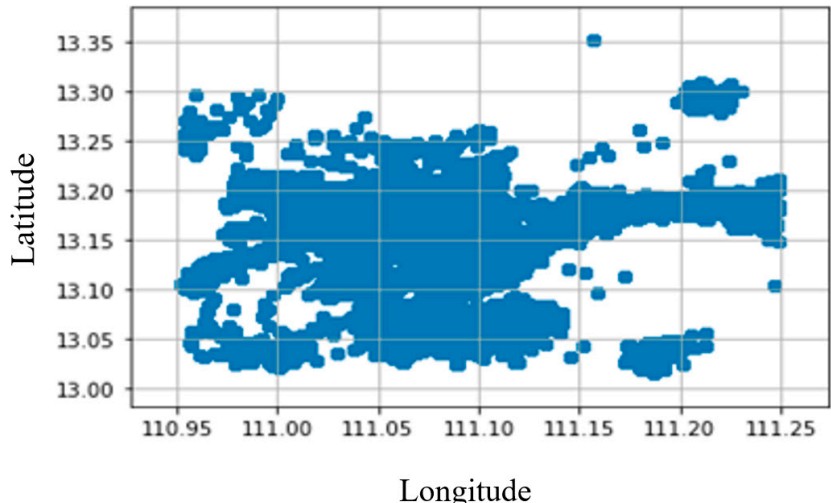

**Figure 7.** Base stations' locations.

To test the performance of this technique, the behavior of base stations belonging to the same location were examined. As illustrated in Figures 8 and 9, base stations at the same location have different traffic loads. The behavior similarity of a group of base stations was clearly not dependent on their location. Therefore, this technique will not be appropriate, and another approach is required to perform the clustering process.

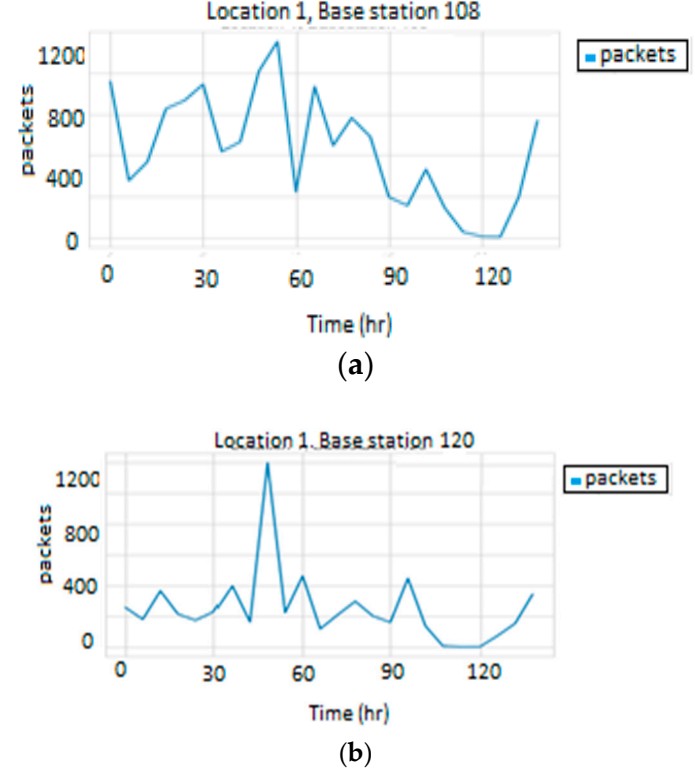

**Figure 8.** Traffic load in Location 1: (**a**) base station 108, (**b**) base station 120.

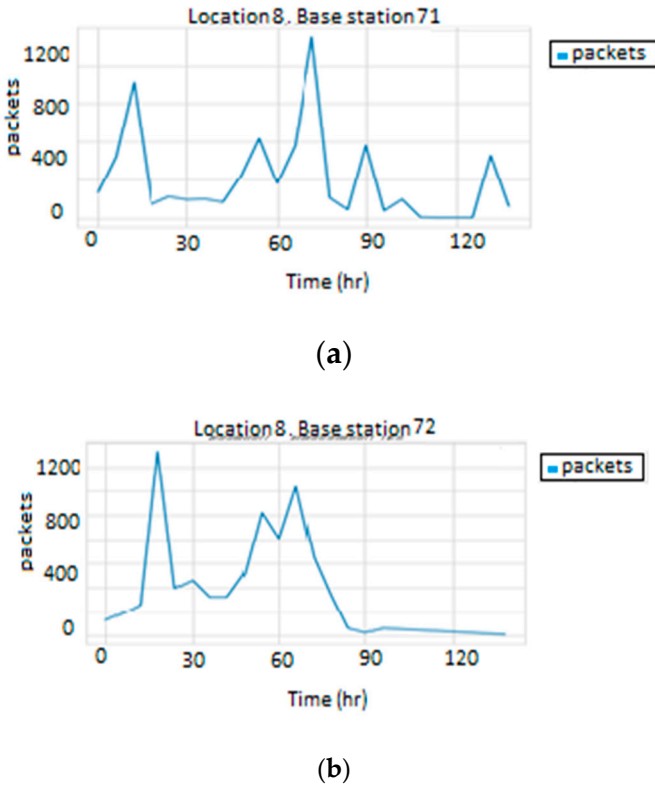

(**a**)

(**b**)

**Figure 9.** Traffic load in Location 8: (**a**) base station 71, (**b**) base station 72.

Instead of the location, time series clustering groups base stations according to their behavior. For instance, as illustrated in Figure 10, although Base Stations 19, 7, 23, and 24 are not necessarily neighbors, they share similar behaviors. Hence, Base Station 19 should be clustered with Base station 7 in Cluster C1, and Base Station 23 should be clustered with Base Station 24 in Cluster C2.

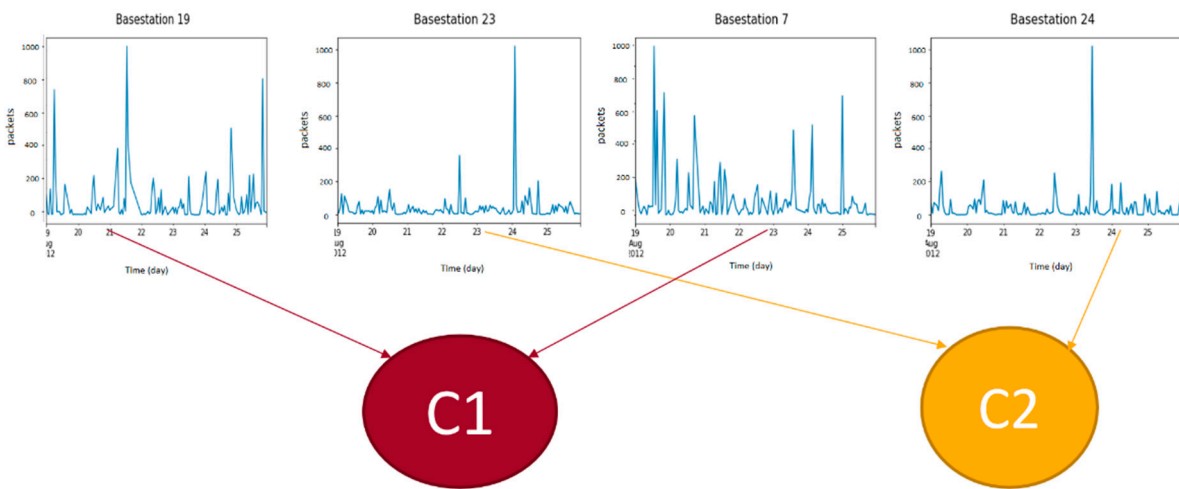

**Figure 10.** Time series clustering.

Now, a metric must be defined to measure the similarity of the base stations. The behavior of a base station can be treated as a time series problem. Several distance measures can be used to measure the similarity between two time series. The second requirement after defining the similarity metric is determining the clustering algorithm by which the base stations are clustered.

In this work, multiple combinations of the most common clustering algorithms and similarity metrics were examined. Global Alignment Kernel KMeans, KShape, and TimeSeriesKMeans were used

with the Euclidean distance and dynamic time warping metrics. The Silhouette index (SIL) was used for the error calculation. The Silhouette index measures the distance between the time series and the centroid of the cluster they belong to compared to the centroids of other clusters. The Silhouette index ranges from −1 to 1. Negative Silhouette values represent incorrect clustering. Values of 0 indicate that the points lie on the decision boundary between different clusters, whereas positive values demonstrate that the points are correctly classified and do not fall on the decision boundary. The closer the Silhouette value to +1, the better the clustering. The Silhouette index equation is defined as ref. [22]:

$$\text{SIL(C)} = \frac{1}{N} \sum_{c_k \epsilon C} \sum_{x_i \epsilon c_k} \frac{b(x_{i,}, c_k) - a(x_i, c_k)}{max\{a(x_i, c_k), b(x_i, c_k)\}} \tag{2}$$

$$\text{a}(x_i, c_k) = \frac{1}{|c_k|} \sum_{x_i \epsilon c_k} d\_e(x_i, x_j) \tag{3}$$

$$\text{b}(x_i, c_k) = min_{c_l \epsilon C \backslash c_k} \{\frac{1}{|c_l|} \sum_{x_i \epsilon c_l} d\_e(x_i, x_j)\} \tag{4}$$

where C represents the cluster's vector, $x$ represents the dataset, $k$ represents the number of clusters, and $d$ represents the distance between the objects.

Different numbers of clusters were attempted in this research; however, the highest scores were produced using five, eight, and ten as the number of clusters. We focused on presenting the results produced using all the explored methods. The results consider the median technique because it is faster than the partition around the medoid. To test the effect of the normalization on clustering, the techniques used were examined using both normalized and non-normalized data.

Tables 1 and 2 provide a comparison between the KShape, Global Alignment Kernel, and TimeSeriesKMeans algorithms. The TimeSeriesKMeans and Global Alignment Kernel algorithms performed better than the KShape algorithm. The reason is that KShape can only deal with single time series representations and cannot operate over multiple representations [23]. Between the TimeSeriesKMeans and Global Alignment Kernel algorithms, TimeSeriesKMeans produced better results of normalized and un-normalized data. This could be due to the high computational complexity of the Global Alignment Kernel algorithm [24], which can be run a few times. A slight increase in the scores can be noted when using eight clusters instead of ten. However, the best results were produced using five clusters. By comparing the results of the normalized data against the non-normalized data, it is revealed that they produce comparable results with a slight increase in the score in the case of the normalized data. The distance metric revealed that using the Euclidean distance resulted in lower scores for both the dynamic time warping (DTW) and Soft-DTW in all the different variations. However, between the dtw and soft-dtw, the results are similar with no noticeable change. With regard to the computation speed, Euclidean distance was very fast compared to both the soft-dtw and dtw, which both required more execution time due to the high time complexity of the dtw algorithm, O($N^2$), compared to O(log(n)) for the Euclidean distance algorithm.

**Table 1.** Results of KShape and global alignment kernel Algorithms.

| Number of Clusters | Silhouette Index KShape Algorithm | | Silhouette Index Global Alignment Kernel Algorithm | |
|:---:|:---:|:---:|:---:|:---:|
| 5 | −0.05 | −0.062 | −0.028 | −0.024 |
| 8 | −0.05 | −0.051 | −0.032 | −0.028 |
| 10 | −0.06 | 0.044 | −0.047 | −0.039 |

**Table 2.** Results of TimeSeriesKMeans algorithm for normalized/un-normalized data.

| Distance Metric | 10 Clusters | | 8 Clusters | | 5 Clusters | |
|---|---|---|---|---|---|---|
| Euclidean | 0.0674 | 0.059 | 0.068 | 0.167 | 0.23 | 0.21 |
| DTW | 0.093 | 0.161 | 0.0957 | 0.188 | 0.39 | 0.35 |
| Soft-DTW | 0.052 | 0.17 | 0.214 | 0.194 | 0.41 | 0.38 |

As discussed above, the partitional clustering method produced the best results. The TimeSeriesKMeans was chosen to be used as the algorithm. Soft dynamic time warping was used as the distance metric, and the number of clusters was set to five. Table 3 presents the number of base stations in each cluster.

**Table 3.** Number of base stations in each cluster.

| Cluster ID | Base Stations Count |
|---|---|
| 0 | 129 |
| 1 | 28 |
| 2 | 1308 |
| 3 | 168 |
| 4 | 438 |

After settling on the clustering technique, a new column was added to the dataset that represents the cluster number for each base station and the base stations were grouped by these cluster numbers.

## 4. Traffic Load Forecasting

In this section, the setup of each model from the examined forecasting models is presented.

### 4.1. Statistical Methods

#### 4.1.1. Auto Regressive Integrated Moving Average (ARIMA) Model

The ARIMA model works with a univariate time series. For testing the statistical models, each base station was trained and tested separately.

The stationarity of a time series is necessary to perform statistical model forecasting. Hence, transformation methods were applied to the data to make it stationary. One characteristic of stationary data is that the values in their autocorrelation function (ACF) plot converge toward zero [25] as the time lag increases. As shown in the ACF plots of some sample base stations in the dataset presented in Figure 11, the data are stationary and do not require further differencing.

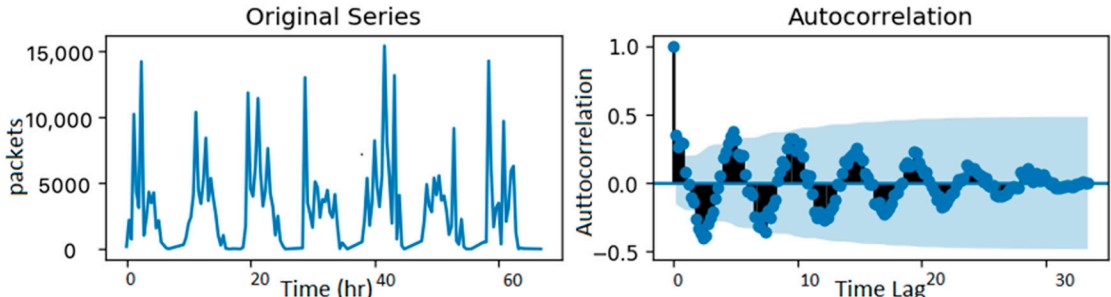

**Figure 11.** Sample cluster autocorrelations.

To set the ARIMA model hyperparameters, $p$, $d$, and $q$, which defines the autoregressive (AR), differencing, moving average (MA) orders, respectively, the grid search method was applied. The search

was in the range of 0 to 3 for the three hyperparameters. The values that produced the least root mean squared error were selected.

### 4.1.2. Seasonal Autoregressive Integrated Moving Average (SARIMA) Model

The SARIMA model is used to handle seasonality in the time series. Figure 12 shows a sample of the decomposition of base stations belonging to each cluster. The seasonal component in each figure demonstrates that there is a seasonality in the data. Hence, SARIMA was applied to explore the performance of handling this seasonality.

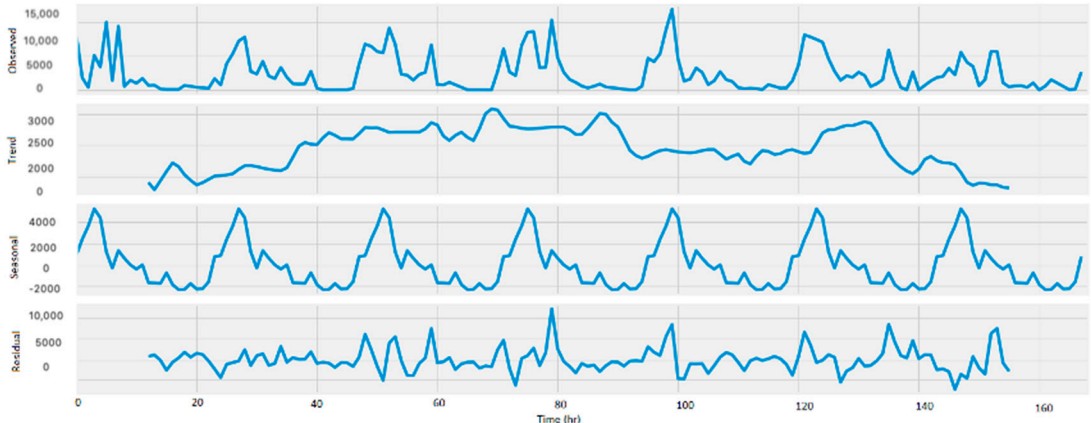

**Figure 12.** Sample cluster decomposition.

SARIMA requires three extra hyperparameters than the ARIMA—P, D, and Q—which define the seasonal AR, seasonal differencing, and seasonal MA orders. As in the ARIMA model, the grid search was applied to find the best hyperparameters of the models, and the root mean squared error was chosen as the error metric.

### 4.2. Machine Learning

In this work, several machine learning techniques are used. A new attribute was added to the dataset to train the machine learning models, representing the traffic load of each base station shifted by the number of steps defined for prediction. This future value was considered as the target for training the models. Each cluster was trained independently of other clusters. Hyperparameter tuning using a grid search was used for selecting the hyperparameters of each algorithm. In our problem, the target value for forecasting is the load value, which is a continuous value and not a discrete one. Therefore, the problem was considered a regression problem. The following sections discuss the examined regression algorithms.

### 4.2.1. Support Vector Machines (SVM)

SVM is a supervised learning algorithm that aims to classify data points by finding the hyperplane that can separate data with minimal error. In the case of nonlinearly separable data, where n-dimensional data cannot be separated using n-1 hyperplane, SVC usually converts data into higher dimensions using the technique kernel trick technique. The kernel trick technique continuously converts the data using a kernel function into higher dimensions until a dimension is found where the data can be separated using a hyperplane. The grid search included three types of kernels: linear, polynomial, and radial basis function (RBF) kernels, which, for two vectors, $x$ and $y$, would have the following formulas:

- Linear Kernel: $k(x, y) = x^T y$;
- Polynomial Kernel: $k(x, y) = \left(1 + x^T y\right)^2$;

- RBF Kernel: $k(x, y) = \exp(-\gamma \|x - y\|^2)$.

$$
\begin{aligned}
\text{Linear Kernel}: \ & k(x, y) = x^T y; \\
\text{Polynomial Kernel}: \ & k(x, y) = \left(1 + x^T y\right)^2; \\
\text{RBF Kernel}: \ & k(x, y) = \exp\left(-\gamma \|x - y\|^2\right).
\end{aligned}
\tag{5}
$$

Between the three choices, the RBF kernel performed better in all models. The kernel was set to RBF in all trained models. For the epsilon, which represents the margin of tolerance that the error should fit in, three values were explored: 0.1, 0.01, and 0.5. For the gamma, which defines the coefficient of the kernel, the search was on 0.1, 0.01, and 0.001. When the value of gamma is small, the constraints on the model will increase and it becomes harder to deal with nonlinear or complex data and vice versa.

### 4.2.2. Multilayer Perceptron (MLP)

MLP is a basic artificial neural network. MLP works by approximating a function f, such that $Y^n = (.) \ X^m$, where m is the number of input features, n is the number of outputs, and * is the application of the function on the input vector. Each hidden layer is represented by h = f (Wx + b), where f is the activation function, W is the set of weights connected to the input, x is the input of the previous layer, and b is the bias. The output is calculated by calculating the result of an activation function on the sum of the product of the weights with the output of the hidden layers. The grid search included four types of hidden layer activation functions: *relu, logistic, tanh*, or *identity* function. For the hidden layers' sizes, (30,30,30), (40,40,40), and (50,50,50) were used. For the alpha, which defines the learning rate, the search was on 0.1, 0.01, and 0.5. The output layer activation function was set as the linear activation function in all models, and the Adam optimizer was used, which calculates adaptive learning rates for each weight individually based on both the mean and variance of the gradients for the weights [20,26].

### 4.2.3. Decision Trees

Decision trees are another supervised learning algorithm that can handle linear and nonlinear problems. Decision trees work by creating a tree-like model of decision questions. Each node represents a question that it will split into another node or provide a result in a leaf. The grid search included the following two types of criteria to calculate the performance of the tree:

- Entropy for information gain;
- Gini for Gini impurity.

These two criteria are used interchangeably; however, based on their implementations, Gini is less computationally intensive. Hence, all the criteria of the models were set to Gini. The splitting parameter was set to "best" for all models, meaning that splitting is made based on the best result instead of randomly. There is no restriction on the depth of the trees.

### 4.2.4. Random Forests

The random forests algorithm is based on the bagging concept, where it is mainly built from several decision trees combined.

The random forests regressor has some similar hyperparameters as the decision trees, such as the criterion and tree max depth. The criterion functions in random forests are either mean squared error or mean absolute error. Another important parameter that should be defined as the number of estimators, which defines how many trees the model contains. The grid search included different numbers of hyperparameters in the range of 10–25. For the criterion function, the mean absolute error (MAE) was selected for all models.

4.2.5. XGBoost

XGBoost is a powerful boosting algorithm that usually outperforms other normal ensemble techniques because, in building new trees, they tend to perform better than previously built trees by avoiding their mistakes. In choosing the hyperparameters, the gamma, which defines the minimum accepted loss for splitting the tree, was chosen to be "0" for all models. For choosing the tree construction method, the tree_method was set to "approx", which uses an approximate greedy algorithm to determine the construction algorithm choice. The search was then on the gbtree, gblinear, and dart boosters. The gbtree booster uses regression trees as weak learners. The dart adds a dropout technique to the gbtree to randomly drop some learners using the dropout rate to prevent overfitting. The gblinear differs from the dart and the gbtree in using linear regression instead of regression trees. The gbtree was chosen because it outperformed the other boosters in all models.

*4.3. Deep Learning*

In this work, deep learning, specifically the recurrent neural network (RNNs) algorithm, was used to examine its performance in predicting the network load on the base stations. RNNs can be defined as neural networks that can capture dependence over time. Because RNNs utilize past patterns, and since time series forecasting depends on the idea that past behavior and patterns can be used to predict future values, recurrent neural networks can be applied in time series forecasting. Traditional RNNs achieve satisfying results when the time dependency is short, meaning that the range of the considered previous time steps is between one and ten steps. If the number of previous time steps increases, the RNN faces the vanishing gradient problem. Many solutions have been suggested to solve the short memory problem in RNNs. In this work, two of the most popular solutions used to solve the short memory problem were examined: LSTMs and GRUs. The novelty of these solutions is that they use gates to learn past sequences that should be kept and those that should be forgotten. By doing so, it can pass important information and track long-term dependencies. Although both LSTMs and GRUs have the same goal of solving the vanishing gradient problem, they differ in their process to reach that goal. The LSTM architecture consists of four gates: forget, learn, remember, and use. The input gate determines what is to be kept of the new cell state, the forget gate determines what to forget from the existing memory, and the output gate decides what the next hidden state should be. The GRU architecture consists of two gates: a reset gate and an update gate. As their names imply, the reset gate determines the past information to be reset or forgotten, and the update gate determines what should be used to update the new cell. For both LSTMs and GRUs, the following parameters should be defined [27]:

- Input dimensions: Defines the input size to the network. For these models, the number in the time series was considered in the training data of each cluster.
- Output dimension: Defines the size of the network output. It considered the number of the time series in each cluster because the resulting prediction is for all inputs.
- Batch size: Defines the amount of data trained in each iteration. In our case, this was "32" for all of the networks.
- Loss function: the root mean squared error was used to measure the loss of the networks.
- Optimizer: The "Adam" optimizer was used as the optimizer in all networks.
- Epochs number: Defines the number of iterations the model is going to run. In our case, 50 epochs were used for all created networks.

For each cluster, one model was created and trained. A sample of base stations belonging to each cluster was considered in the results, and their traffic loads were shown against the forecasting. The root mean square error (RMSE), which was calculated as the average RMSE for all base stations belonging to the same cluster, was presented.

Regarding the architecture of the networks, the LSTM networks included two LSTM layers: a dropout layer to prevent overfitting and a fully connected network layer. The same architecture for the GRU networks was used, except GRUs used network layers instead of the LSTM layers.

### 4.4. Unclustered Data

To examine the effect of clustering on forecasting, the network ran on the complete dataset. The dataset was split into 70% for training, 10% for validation, and 20% for testing. The forecasting was tested then using the same techniques used on the clustered version of the data.

### 4.5. AWS DeepAR Algorithm

DeepAR is among the algorithms offered by SageMaker [28]. It is a supervised learning algorithm designed for time series forecasting using autoregressive recurrent neural networks (RNNs). The main advantage of the DeepAR algorithm is that it utilizes the similarity found between related time series data to improve forecasting performance by learning a global model from the past data of the existing related time series. DeepAR inventors claim that DeepAR outperforms traditional forecasting methods in the existence of a large number of related time series. They have also mentioned that because the model learns from related time series, DeepAR can provide future predictions for new time series with no previous records [29].

DeepAR requires the data to be in a specific JSON format and uploaded to SageMaker. Data preprocessing was performed to be able to use the model. In Table 4, the hyperparameters that were defined to train the DeepAR model are presented [30].

**Table 4.** Hyperparameters values for the DeepAR model.

| Hyperparameters | Value |
|---|---|
| Time frequency | hourly |
| Epochs | 100 |
| Loss | RMSE |
| Number of Layers | 3 |
| Learning Rate | 0.0001 |
| Batch Size | 32 |
| Number of cells | 50 |
| Likelihood | Gaussian |
| Dropout Rate | 0.05 |

## 5. Results

Various loss functions can be used for regression problems, such as mean absolute error (MAE), mean squared error (MSE), and root mean square error (RMSE), calculated as follows.

$$(\text{MSE}): \text{Loss} = \frac{1}{N}\sum_{i=1}^{N}(y_i - \hat{y})^2 \tag{6}$$

$$(\text{RMSE}): \text{Loss} = \sqrt{\frac{1}{N}\sum_{i=1}^{N}(y_i - \hat{y})^2} \tag{7}$$

$$(\text{MAE}): \text{Loss} = \frac{1}{N}\sum_{i=1}^{N}|y_i - \hat{y}| \tag{8}$$

where $N$ is the total error number, $\hat{y}$ is the prediction value, and $y_i$ is the true value.

The equations above show that both MSE and RMSE calculate the square of the error before averaging; accordingly, larger errors provide relatively larger weights. Hence, RMSE and MSE penalize

large errors. As RMSE takes the square root of MSE, it is calculated in the same units of the actual observations, making it more comprehensible than MSE. Therefore, in this paper, RMSE was used as the error metric for measuring the performance of the different models.

In this section, the RMSEs resulting from each model are provided. Two sample plots of representative resulting base stations are presented under each table. The plots show the actual traffic on the base station against the resulting forecasting.

## 6. Discussions

The presented results in Tables 5 and 6 and Figures 13 and 14 reveal that the best performance produced was by the recurrent neural networks. The results provided by the GRU and LSTM networks were close to each other, albeit with a slight decrease in the RMSE for the GRU networks. The resulting error using the clustered version of the data was less than the error produced with no clustering, showing that the network performs better when the similarity of behavior among the training data increases. Compared to the error generated by the available online tool DeepAR, the error produced from our created models was close. It should be noted that the reported errors using the machine learning and statistical models were higher than those produced using the recurrent neural networks. The statistical methods' error was less than the error produced using the machine learning algorithms, confirming that statistical methods are better than machine learning for small data [31]. Among the different machine learning algorithms explored, the MLP algorithm resulted in the lowest error in most clusters. The decision trees algorithm produced the highest error for most clusters. The poor performance of the decision trees supports the earlier belief that decision trees are not as useful in regression as they are in classification [32]. The best performance among all used techniques was achieved by the recurrent neural networks. In contrast, the traditional artificial neural networks using MLP did not perform as well, highlighting the impact of memory in neural networks on handling time series forecasting tasks. The performance of the SARIMA model is better than the ARIMA model, most likely because of the seasonality found in the data. To conclude, the following observations can be made:

1. Among the different forecasting algorithms, the recurrent neural networks perform best in the time series forecasting problem at the exiting of multiple time series.
2. Clustering can significantly improve forecasting performance because it increases the consistency of the training data.
3. XGBoost does not always provide better performance than other machine learning models.
4. In the case of a small amount of data, the statistical models may result in better forecasting than the machine learning models.

**Table 5.** RMSE for all models. Performance based on clustered data.

| Cluster ID | ARIMA | SARIMA | SVM | MLP | Decision Trees | Random Forests | XGBoost | LSTM | GRU |
|---|---|---|---|---|---|---|---|---|---|
| Cluster 0 | 2759 | 1013 | 3118 | 2902 | 3004 | 3000 | 3520 | 1532 | 1205 |
| Cluster 1 | 2120 | 1215 | 2584 | 2009 | 3321 | 2172 | 2416 | 1140 | 982 |
| Cluster 2 | 1253 | 3840 | 1474 | 1265 | 1531 | 1379 | 1147 | 690 | 451 |
| Cluster 3 | 1591 | 1075 | 2326 | 1708 | 2427 | 2464 | 2605 | 1097 | 724 |
| Cluster 4 | 2759 | 1038 | 1587 | 795 | 1635 | 1023 | 1314 | 1311 | 1113 |

**Table 6.** Unclustered data and DeepAR RMSE.

| Unclustered Data | DeepAR |
|---|---|
| 1562 | 1206 |

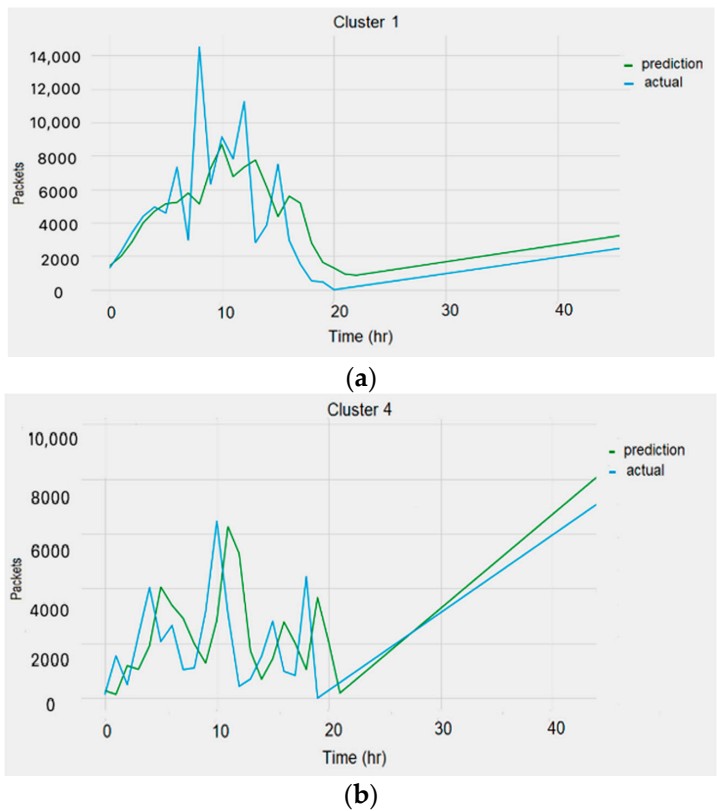

**Figure 13.** Sample plots: (**a**) Cluster 1, (**b**) Cluster 4.

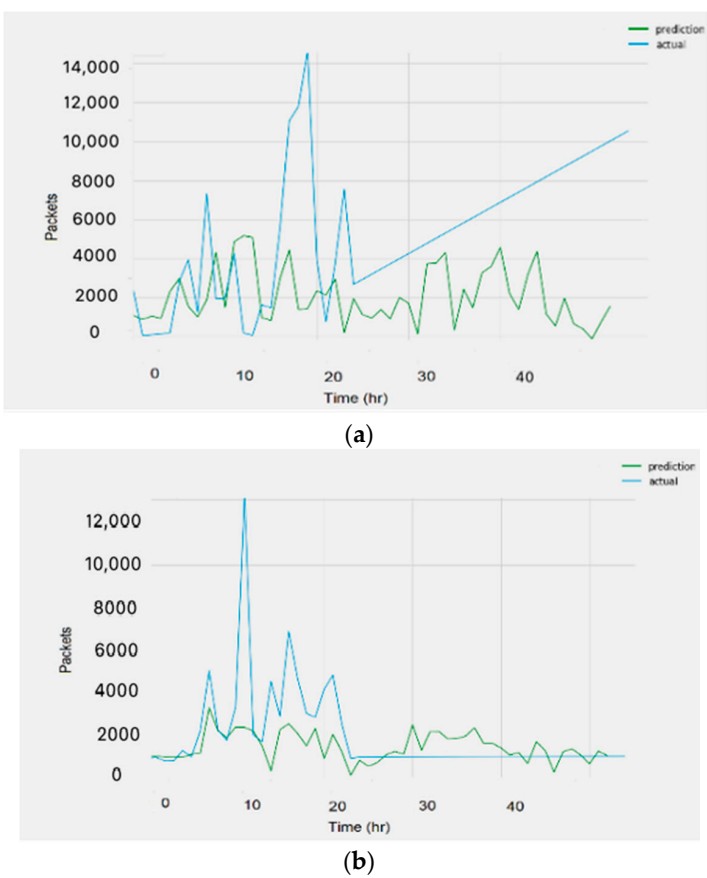

**Figure 14.** Unclustered data sample plots. (**a**) Cluster 1, (**b**) Cluster 4.

## 7. Conclusions

In this research, network traffic load prediction on base stations was modeled as a time series forecasting problem. The city cellular traffic dataset, a public dataset that provides traffic load statistics for 13 k base stations in China, was used for our work. Clustering according to behavior was proposed as a solution for the nonsimilar behavior of different base stations. Different clustering techniques with different distance metrics were explored to acquire the techniques and number of clusters that provide the least error. The least error resulted from using Timeseries KMeans as the clustering algorithm with Soft-DTW as the distance metric. After trying different numbers of clusters, the base stations were clustered into five clusters and grouped based on the cluster number. This step was performed to train and test each group separately. Statistical methods, machine learning, and recurrent neural networks, commonly used in time series forecasting techniques, were examined on the dataset. The performance of the algorithms was tested by the resulting root mean square error. RNNs showed better performance than both the statistical methods and machine learning algorithms. The results suggest that a clustered base station is better than using all of the base stations. Amazon DeepAR, an online tool built for the time series forecasting problem, was utilized as a benchmark to compare the tested algorithms. The findings showed similar error values for the two groups.

The main conclusions of this research can be summarized in the following remarks:

1.  Clustering base stations based on their behavior before using the time series forecasting methods to predict their loads can significantly improve the accuracy of their load forecasting when an adequate number of records exists in each cluster.
2.  Several clustering algorithms can be used for clustering time series data. Among them, TimeSeriesKMeans usually results in the lowest error.
3.  Recurrent neural networks, such as LSTMs and GRUs, usually provide better performance than statistical methods and machine learning for time series forecasting.

As previously mentioned, the dataset used in this work has several limitations. Future work must consider applying and testing the techniques used in this work with other datasets that overcome these limitations. Another issue that can be investigated in future work is the forecasting horizon. Because multistep forecasting increases uncertainty, leading to an increase in error, this research focused on single-step forecasting. Therefore, for future work, models could be examined on higher forecasting horizons.

**Author Contributions:** Conceptualization, H.S.H., H.A., A.N. and B.M.; Methodology, H.A., H.S.H. and B.M.; Software, B.M.; Validation, H.A. and B.M.; Formal Analysis, H.A., H.S.H., and B.M.; Investigation, H.A., H.S.H. and B.M.; Resources, H.S.H., A.N., and H.A.-z.; Writing—Original Draft Preparation, B.M., and H.A.; Writing—Review & Editing, H.A., H.S.H., A.N., and H.A.-z.; Visualization, H.A., and B.M.; Supervision, H.A., H.S.H., A.N. and H.A.-z.; Project Administration, H.A., H.S.H., A.N., and H.A.-z.; Funding Acquisition, H.S.H., and A.N. All authors have read and agreed to the published version of the manuscript.

**Funding:** This research is supported by a grant from the Natural Sciences and Engineering Research Council of Canada (NSERC) under Grant Number STPGP 521432-2018.

**Conflicts of Interest:** The authors declare no conflict of interest.

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
