# Peer review of "A Clustering-Driven Approach to Predict the Traffic Load of Mobile Networks for the Analysis of Base Stations Deployment"

_jsan, doi:10.3390/jsan9040053_

Round 1

Reviewer 1 Report

The following issues need to be solved or clarified to improve the quality of the paper: 

  1. Page 1, Introduction, Line 2: data traffic → mobile data traffic
  2. Page 2, the authors claimed, "With the ability to predict the traffic on base stations, the opportunity to save energy arises by developing a method to switch off base stations during the off-load times." Please consider adding more explanation on the use cases of your proposed methods, as switching off base stations may have side effects (such as low coverage rate and service level) and current network planning and deployment have already considering the energy consumption issue. Also, please consider adding 1-2 paragraphs in the literature part on how 5G networks solve these issues (5G is one of the keywords of the paper, but the content and data used for building models seem to have no relevance to 5G). 
  3. Resolutions of Figures: please replace all the low-resolution figures in the paper, since the legends and data points in some figures can not be read easily and clearly.
  4. Page 6: is there any root symbol outside St and St-24?
  5. Page 8: in Table 1, when the cluster number is 10 and using the non-normalized data, the Silhouette Index is 0.044 and seems different from the trends of other results. Please explain this.

  6. Page 8: Table 2: 0 clusters should be 10 clusters?; Also, when under 5 clusters and SoftDTW, the Silhouette Index results are 0.41 and 0.38. The results are better than others, but I am not sure whether the clustering results are satisfactory since is there is no threshold value for reference. Please explain the results and show why clustering results are sufficient for being used in the next step (prediction).
  7. Page 10: figure numbering issue: you lost Figure 13 between 12 and 14.
  8. In Section 4 Results, RMSE is used as a metric for measuring the performance of models. Please explain the reason for using this metric instead of using others such as MSE and MAE.

Reviewer 2 Report

This paper adequately put the progress it reports in the context of previous work, representative referencing, and introductory discussion. It is clearly and concisely written. The conclusions and potential impacts of the paper are made clear.
I would suggest the following to the authors:

  • Indeed, the authors referred to another reference [2] showing the benefits from the prediction, however, in this paper the motivation/significance from the prediction is not clear.
  • The optimality of the model that performs well in detecting spike is not well-established mathematically. It is recommended to make claims only when they are mathematically justified, which I believe is not necessary for this work.
  • Threshold of 67 which represents a value of 40% of the maximum number of records (24 hours *7 days = 168 hours) was set. Only base stations with the number of records exceeding this threshold (67) are kept”-->Authors can justify why 67. Also, the traffic pattern fluctuates over both time and space, in which the under-utilized BSs can be useful in the framework.
  • The Z-score of all the data points were calculated. Then a threshold of three was determined.”-->Authors can justify the 3 how it is determined
  • Soft-Dynamic Time Warping was used as the distance metric and the number of clusters was set to five” -->authors can justify why 5.
  • For the haversine distance, authors can add boundaries to the equation to ensure that what is under the root does not exceed 1
  • “The TimeSeriesKMeans produced better results in both forms of normalized and un-normalized data than the KShape and the Global Alignment Kernel algorithms.”-->if possible, the authors can explain why the time series KMeans produces better results.

Minor:

  • The x and y-axis in all figures are not clear
  • Proofreading is advised, the word “project” repeated a couple of times, should be replaced by paper.
